

# Salivary interleukin-17A and interleukin-18 levels in patients with celiac disease and periodontitis

Marwa Madi[1], Maha Abdelsalam[2], Ahmed Elakel[1], Osama Zakaria[2], Maher AlGhamdi[3], Mohammed Alqahtani[3], Luba AlMuhaish[3], Faraz Farooqi[4], Turki A. Alamri[5], Ibrahim A. Alhafid[5], Ibrahim M. Alzahrani[5], Adel H. Alam[5], Majed T. Alhashmi[5], Ibrahim A. Alasseri[5], Ahmad A. AlQuorain[6] and Abdulaziz A. AlQuorain[5]

[1] Department of Preventive Dental Sciences, College of Dentistry, Imam Abdulrahman Bin Faisal University, Dammam, Eastern Province, Saudi Arabia
[2] Department of Biomedical Dental Sciences, College of Dentistry, Imam Abdulrahman Bin Faisal University, Dammam, Eastern Province, Saudi Arabia
[3] College of Dentistry, Imam Abdulrahman Bin Faisal University, Dammam, Eastern Province, Saudi Arabia
[4] Department of Dental Education, College of Dentistry, Imam Abdulrahman Bin Faisal University, Dammam, Eastern Province, Saudi Arabia
[5] Department of Internal Medicine, Gastroenterology Division, King Fahad University Hospital, Imam Abdulrahman Bin Faisal University, Dammam, Eastern Province, Saudi Arabia
[6] College of medicine, Imam Abdulrahman Bin Faisal University, Dammam, Eastern Province, Saudi Arabia

Corresponding author
Marwa Madi, mimadi@iau.edu.sa

## ABSTRACT

**Background:** An increased level of interleukin-17A and interleukin-18 in the serum and intestinal mucosa of celiac disease patients reflecting the severity of villous atrophy and inflammation was documented. Thus, the objective of this study was to evaluate the concentrations of salivary-17A, interleukin-1 beta, and interleukin-18 in patients with celiac disease who are on a gluten-free diet, both with and without periodontitis, and to compare these levels with those in healthy individuals.

**Methods:** The study involved 23 participants with serologically confirmed celiac disease (CD) and 23 control subjects. The CD patients had been following a gluten-free diet (GFD) for a minimum of 1 year and had no other autoimmune disorders. The research involved collecting demographic data, conducting periodontal examinations, gathering unstimulated whole saliva, and performing enzyme-linked immunosorbent assays to measure salivary interleukin-17A, interleukin-1 beta, and interleukin-18 levels. Spearman's correlation analysis was utilized to explore the relationships between CD markers in patients on a GFD and their periodontal clinical findings.

**Results:** The periodontal findings indicated significantly lower values in celiac disease patients adhering to a gluten-free diet compared to control subjects ($p = 0.001$). No significant differences were found in salivary IL-17A, IL-18, and IL-1B levels between celiac disease patients and control subjects. Nevertheless, the levels of all interleukins were elevated in periodontitis patients in both the celiac and control groups. The IL-1 Beta level was significantly higher in periodontitis patients compared to non-periodontitis patients in the control group ($p = 0.035$). Significant

negative correlations were observed between serum IgA levels and plaque index (r = −0.460, $p$ = 0.010), as well as gingival index (r = −0.396, $p$ = 0.030) in CD patients on a gluten-free diet.

**Conclusion:** Celiac disease patients on gluten-free diet exhibited better periodontal health compared to control subjects. However, increased levels of salivary IL-17A, IL-18 and IL-1B levels were associated with periodontitis. Additionally, serum IgA level was significantly inversely associated with periodontitis clinical manifestations and with salivary inflammatory mediators in CD patients on GFD.

## INTRODUCTION

Celiac disease is a chronic autoimmune illness that affects the small intestine and is caused by the consumption of gluten (*de Lima et al., 2016*; *Spinell et al., 2018*). It is characterized by intestinal inflammation, villous atrophy, malabsorption and increased risk of other autoimmune diseases (*Mazurek-Mochol et al., 2021*).

The global prevalence of celiac disease was reported to be 1.4% in almost 300,000 individuals, based on positive results from tests for anti–tissue transglutaminase and/or anti-endomysial antibodies (*Singh et al., 2018*). The only efficient therapy for celiac disease is gluten-free diet for life, which can improve intestinal mucosal damage and reduce the symptoms and complications of the disease (*Spinell et al., 2018*).

Periodontitis is a systemic inflammation disease that damages the teeth's supporting tissues and causes tooth loss. It is caused by the interaction of oral bacteria and host immune response (*Barada et al., 2012*; *Emampanahi et al., 2019*). Periodontitis arises when the balance between bacterial biofilm and host responses is disrupted, often due to dysbiosis or an exaggerated immune response. This complex interplay, influenced by variations in dental plaque, genetics, and immune profiles, culminates in an intensified inflammatory state, ultimately causing the tissue damage characteristic of periodontal disease (*Hajishengallis & Chavakis, 2021*; *Kinane, Stathopoulou & Papapanou, 2017*).

Diabetes, cardiovascular disease, rheumatoid arthritis, and inflammatory bowel illness have all been linked to periodontitis (*Holmstrup et al., 2017*; *Madi et al., 2021*). The prevalence of periodontitis varies from 10% to 90% in different populations and regions. The main treatment for periodontitis is mechanical removal of plaque and calculus, which can reduce the bacterial load and inflammation (*Mokeem et al., 2018*).

IL-1β is crucial in developing trained immunity (*Hajishengallis & Chavakis, 2021*). Animal studies indicated that periodontal pathogens can exacerbate intestinal inflammation by affecting gut dysbiosis and barrier integrity, suggesting a mutual relationship between oral and gut health (*Kitamoto et al., 2020*). Notably, certain oral pathogens like Klebsiella spp. and Enterobacter spp. can aberrantly settle in the gut, inciting colitis by triggering non-canonical IL-1β secretion from inflammatory macrophages (*Hajishengallis & Chavakis, 2021*). IL-1β significantly contributes to the

destruction of periodontal tissues by enhancing bone resorption and increasing the production of enzymes that degrade tissue. Moreover, along with other pro-inflammatory cytokines like TNF-α and IL-6, IL-1β influences not just periodontitis but also systemic diseases, highlighting its role in broader inflammatory pathways (*Cardoso, Reis & Manzanares-Céspedes, 2018*; *Cheng et al., 2020*).

CD arises in patients with HLA-DQ2 or HLA-DQ8 alleles and is characterized by a T-cell-driven inflammation in the proximal small bowel activated by gluten ingestion (*Ashtari et al., 2019*). Gluten then goes to TCD4+ cell and stimulate the secretion of different pro-inflammatory and inflammatory cytokines like interferon (IFN)-γ, interleukin IL-1, IL-18, IL17, and IL-21 genes (*Nasserinejad et al., 2019*). This will lead to interaction between TCD4+ with B cells and the release of autoantibodies such as anti-tissue transglutaminase (tTG) and DGA (deamidated gliadin antibody) which is a marker for active disease (*Nasserinejad et al., 2019*).

Interleukin-17A (IL-17A) and interleukin-18 (IL-18) are pro-inflammatory cytokines that play important roles in both celiac disease and periodontitis (*Diaz-Castro et al., 2020*; *Techatanawat et al., 2020*). IL-17A is produced by Th17 cells, a subset of CD4+ T cells that are involved in mucosal immunity and tissue inflammation (*Rodrigo et al., 2018*). IL-17A stimulates the production of other inflammatory mediators and chemokines, enhances neutrophil recruitment and activation, and promotes tissue destruction (*Mazurek-Mochol et al., 2021*).

IL-18 is produced by macrophages, dendritic cells and epithelial cells, and acts as a co-stimulator of Th1 cells, which produce interferon-gamma (IFN-γ) (*Zhong, 2014*). IL-18 also enhances the production of IL-17A by Th17 cells (*Liukkonen et al., 2016*). Previous studies (*Mazurek-Mochol et al., 2021*; *Pietz et al., 2017*) have shown that IL-17A and IL-18 are elevated in the serum and intestinal mucosa of celiac disease patients, and correlate with the severity of villous atrophy and inflammation.

IL-17A and IL-18 levels have been shown to be elevated in the serum and gingival crevicular fluid of periodontitis patients as well as being correlated with periodontal disease clinical manifestations severity (*Borilova Linhartova et al., 2016*; *Techatanawat et al., 2020*; *Zhong, 2014*).

However, there is limited information on the levels of IL-17A and IL-18 in saliva, which is a non-invasive biological fluid that reflects both systemic and oral health status (*Borilova Linhartova et al., 2016*).

As a result, the study's aim was to investigate the levels of salivary IL-17A, IL-18, and IL-1B in celiac disease patients on GFD with and without periodontitis compared to healthy controls. We also evaluated the associations between these cytokines and celiac disease markers, such as serum IgA level, anti-tissue transglutaminase antibody (tTG) level, and periodontal clinical findings, such as plaque index (PI), gingival index (GI), bleeding upon probing (BOP), probing pocket depth (PPD) and clinical attachment loss (CAL).

## MATERIALS AND METHODS

The guidelines for STROBE were utilized to guide the reporting for this observational cross-sectional study. The Institutional Review board of Imam Abdulrahman Bin Faisal
University IAU approved the study proposal (IRB-2021-02-034), and all subjects provided informed consent. The study was conducted following the Declaration of Helsinki.

In the current study, 23 (older than 18-year) CD patients confirmed by serological tests serum Transglutaminase tTG IgA and antibody (tTGA), Anti_gliadin_Ab, and/or anti-endomysial antibody (EMA IgA) were recruited to the celiac clinic at King Fahd University Hospital Gastroenterology division for routine celiac disease checks (diet, symptoms check, serological tests).

The detection of serum tTGIgA, tissue Transglutaminase antibody (tTGA) and Anti_gliadin_Ab was conducted using the enzyme-linked immunosorbent assay (ELISA) method, employing kits provided by INOVA Diagnostics Inc., San Diego, CA, USA, in strict accordance with the manufacturer's protocols. Venous blood was drawn and centrifuged to separate the serum. The concentration of total serum IgA was determined through antigen-antibody interactions within the serum samples. This process involves the formation of immune complexes, leading to an increase in light scattering, which is then measured and quantified.

For tTGA antibody detection, serum samples were diluted at a 1:100 ratio with distilled water and incubated with recombinant human tTG antigen for 30 min at room temperature. After incubation, the samples were washed three times and subsequently incubated for another 30 min at room temperature with anti-human IgA. The optical density of the resultant solution was measured at 450 nm, and the results were calibrated against a reference standard from the manufacturer.

For the detection of EMA IgA levels, an indirect immunofluorescence assay was employed. Serum samples were diluted at a 1:5 ratio with phosphate-buffered saline and then applied to 4-mm-thick cryosections of distal monkey oesophagus tissue. The samples were incubated for 30 min, followed by the addition of fluorescein isothiocyanate-labelled rabbit anti-human IgA (Biosystems, Barcelona, Spain) to detect the primary antigen-antibody reaction. A positive result was indicated by the emission of bright green reticular fluorescence.

Inclusion criteria were patients having celiac disease with no other systemic condition and on gluten-free diets for at least 1 year. The control group consisted of 23 people seeking standard dental care who had no history of celiac disease or other autoimmune illnesses. They underwent annual checkups without complicating risk factors. The exclusion criteria were pregnancy, current breastfeeding, ulcerative colitis, presence of other autoimmune disorders, and previous head and neck radiotherapy.

## Clinical examination

Patient's gender, age, and the duration of being on gluten free diet was collected. Gingival index GI, bleeding on probing BOP using a blunt probe plaque index PI, Probing depth PD and attachment loss (CAL) were collected. The clinical examination was carried out by two pre-calibrated examiners utilizing a periodontal probe (UNC-15 probe, Hu-Friedy, Chicago, IL, USA). A recorder entered clinical data into forms designed expressly for the study. Clinical assessments such as probing depth PD and clinical attachment loss CAL were performed twice on ten patients within a week to measure intra-examiner reliability.

Minor variations were noticed, however, all second measurements agreed with the initial readings (Kappa 0.9).

A standardized examiner did a thorough periodontal assessment (intraclass correlation coefficient >0.9) with graduated periodontal probe (UNC-15; Hu-Friedy, Chicago, IL, USA). For all teeth except third molars, the following clinical factors were documented at six sites per tooth (mesiobuccal, midbuccal, distobuccal, mesiolingual, midlingual and distolingual): PI (*Silness & Löe, 1964*), GI (*Löe & Silness, 1963*), PPD (distance between the gingival side and the end of the pocket), and CAL (distance between the cementoenamel junction and the bottom of the pocket). A diagnosis of periodontitis is established by two or more interproximal sites with CAL >3 mm and PPD >4 mm (not on the same tooth) or at least one interproximal site with PPD >5 mm (*Tonetti, Greenwell & Kornman, 2018*).

## Protocol for estimation of salivary inflammatory biomarkers

After an overnight fast, a saliva sample was taken between 9:00 AM and 12:00 PM on the exact same day. Whole saliva was collected using spitting method (*Fey, Bikker & Hesse, 2024*; *Navazesh, 1993*; *Navazesh & Kumar, 2008*) so as to collect both stimulated and unstimulated saliva. A sterile tube was placed on ice and the subjects were instructed to spit approximately 5 ml of saliva. After saliva collection, a protease inhibitor cocktail (Roche Diagnostics GmbH, Mannheim, Germany) was added. A centrifugation of the saliva at 800 g at 4 °C for 10 min was followed by the collection of the supernatant, which was aliquoted and stored at −80 °C.

## IL-17A, IL-18, and IL- 1B Measurements

The quantities of IL-17A, IL-18, and IL-1B in saliva samples were determined using a commercially available enzyme-linked immunosorbent assay (ELISA) (xMark, BIO-RAD, Microplate Spectrophotometer, Santa Clara, CA, USA) in 96-well plates (Nunc Maxisorp, Dominique Dutscher) using commercially available human ELISA kits (MOLEQULE-ON, Auckland, New Zealand).

All experiments were performed in triplicate and the procedures were conducted according to the manufacturer's instructions. The sandwich ELISA principle was used in all of the kits. The samples were liquefied at room temperature and handled according to the manufacturer's instructions. For the ELISA tests, 100 µL of saliva samples were diluted 1:2. In short, every sample received of 100 µL underwent incubation for 120 min at 25 °C (IL-17A) and 90 min at 37 °C (IL-18 and IL-1B). After washing steps, 100 µL of Biotin-conjugated anti- human IL-17 antibody solution was added into each well and incubated for 1 hr at 25 °C (for IL-17A) and 37 °C (for IL-18 and IL-1B). After washing away any excess antibody, 100 µL of streptavidin-HRP conjugate was added and incubated for 30 min at 25 °C (for IL-17A) and 37 °C (for IL-18 and IL-1B). 100 µL of substrate solution underwent incubation for 15 min in the dark at 25 °C (for IL-17A) and 37 °C (for IL-18 and IL-1B). Using standard curves, the levels of IL-17A and IL-18 in each sample were determined and expressed in pg/mL. The lower detection limits for IL-17A, IL-18, and IL-1B were 8.17, 2.8, and 1 pg/mL, respectively.

**Table 1 Demographic data.**

| Demographics | | Control n (%) | Celiac n (%) |
|---|---|---|---|
| Gender | Male | 10 (43) | 3 (13) |
| | Female | 13 (57) | 20 (87) |
| Disease | Periodontitis | 19 (83) | 7 (30) |
| | Non-periodontitis | 4 (17) | 16 (69) |

## Statistical analysis

Descriptive statistics were computed using mean, standard deviation, frequency, and percentages where appropriate. Normality of data was checked using Shaprio-Wilk's test and $p$-values less than 0.05 showed significant deviation of data from normal distribution. The comparison between salivary biomarkers and periodontal indicators was carried out using the Mann Whitney test. The correlation between clinical parameters, biomarkers and Serological diagnosis was calculated by Spearman's correlation coefficient. Multiple linear regression models (separate models) were developed in order to investigate the association of all biomarkers as dependent variables with the disease (Celia $vs$ Control) as independent variable. $p$-values less than or equal to 0.05 was considered as statistically significant. Statistical package for social sciences (SPSS, Inc IBM, Armonk, NY, USA) version 24 was used for statistical analysis.

## RESULTS

This study involved 46 participants, divided into two groups: control (23) and celiac (23). The control group consisted of 13 (57%) females, averaging 41 years old, while the celiac group comprised 26 (87%) female patients. All participants were nonsmokers. When categorized by periodontal disease status, 83% of the control group and 30% of the celiac group were diagnosed with periodontitis. The demographic characteristics of both groups are presented in Table 1.

Periodontal parameters, including the gingival index (GI), bleeding on probing (BOP), minimum and maximum probing depth (PD), and clinical attachment loss (CAL), were significantly higher in the control group compared to the celiac group ($p < 0.001$), as shown in Table 2.

The salivary biomarkers IL-1 Beta, IL-17A, and IL-18 showed non-significant elevated values in the control group than in the celiac group (Table 3). In the control group, intra group comparison showed that periodontitis patient had higher PI scores, GI, and BOP compared to healthy individuals, with significant differences in GI and BOP ($p = 0.007$, $p = 0.001$, respectively). Salivary IL-1Beta levels were notably higher in periodontitis patients compared to healthy individuals, reaching statistical significance ($p = 0.035$) (Table 4). In the celiac group, intra group comparison showed that while PI scores, GI, BOP, and salivary IL levels were higher in periodontitis patients, these differences did not reach statistical significance. Notably, BOP was significantly higher in periodontitis patients ($p = 0.007$) within the celiac group (Table 4). Periodontitis patients in the control

**Table 2 Clinical periodontal findings in the examined groups.**

| Periodontal findings | | N | Mean | Std. Deviation | *p*-value |
|---|---|---|---|---|---|
| GI Score | Celiac | 23 | 1.24 | 0.37 | 0.001* |
| | Control | 23 | 1.7 | 0.47 | |
| BOP (%) | Celiac | 23 | 0.1 | 0.14 | 0.001* |
| | Control | 23 | 0.29 | 0.13 | |
| min_PD (mm) | Celiac | 23 | 1.30 | 0.47 | 0.001* |
| | Control | 23 | 2.52 | 0.73 | |
| max_PD (mm) | Celiac | 23 | 3.00 | 0.85 | 0.001* |
| | Control | 23 | 4.43 | 0.90 | |
| min_CAL (mm) | Celiac | 23 | 0.26 | 0.45 | 0.001* |
| | Control | 23 | 1.39 | 0.94 | |
| max_CAL (mm) | Celiac | 23 | 0.39 | 0.78 | 0.001* |
| | Control | 23 | 3.00 | 2.00 | |

**Note:**
* Significant level at *p* < 0.05.

**Table 3 Mean values of salivary IL-17A, IL-18, and IL_1Beta in pg/mL.**

| Biomarkers | | N | Mean | Std. Deviation | *p*-value |
|---|---|---|---|---|---|
| IL_17A | Celiac | 23 | 2.020 | 1.887 | 0.929 |
| | Control | 23 | 2.06 | 1.56 | |
| IL_18 | Celiac | 23 | 350.1 | 204.53 | 0.588 |
| | Control | 23 | 384.42 | 222.07 | |
| IL_1Beta | Celiac | 23 | 81.85 | 85.04 | 0.971 |
| | Control | 23 | 82.77 | 84 | |

group showed significantly higher GI ($p = 0.007$) and BOP ($p = 0.001$) than those in the celiac group.

Table 5 displays the association between clinical parameters, biomarkers, and serological diagnosis. Spearman's correlation analysis revealed that the Plaque index (PI) was statistically negatively correlated with IgA ($r = -0.520$, $p = 0.011$). Similarly, GI and BOP were inversely and significantly correlated with IgA ($r = -0.584$, $-0.594$, $p = 0.003$, respectively). Salivary biomarkers did not exhibit any significant or strong association with serological diagnosis tests. IL-17 showed a negative correlation with tTGA, IgA, and IgG, while IL-18 was negatively correlated with EMA and IgA. IL-1B showed a negative correlation with IgA.

Multiple linear regression models were developed to investigate the association of all biomarkers with the disease. Table 6 presents the coefficients of linear regression. All periodontal biomarkers, including GI score, minimum PD, maximum PD, minimum CAL, maximum CAL, and BOP, showed a significant but inverse association with the disease (Control *vs* Celiac). Conversely, salivary biomarkers did not demonstrate a significant association with the disease (Table 6).

**Table 4 Salivary biomarkers' level and periodontal parameters in periodontitis and non-periodontitis patients in the control and celiac groups.**

| Markers | Control Group | | Celiac Group | | p-value [a] | p-value [b] |
|---|---|---|---|---|---|---|
| | Periodontal disease [a] | Healthy [b] | Periodontal disease [a] | Healthy [b] | | |
| PI | 1.38 + 0.27 | 1.02 + 0.05 | 1.28 + 0.37 | 1.01 + 0.02 | 0.157 | 0.536 |
| p-value | 0.214 | | 0.134 | | | |
| GI | 1.852 + 0.362 | 1.167 + 0.408 | 1.32 + 0.41 | 1.03 + 0.11 | 0.007* | 0.297 |
| p-value | 0.007* | | 0.999 | | | |
| BOP (%) | 0.336 + 0.095 | 0.048 + 0.027 | 0.16 + 0.16 | 0.02 + 0.04 | 0.001* | 0.194 |
| p-value | 0.001* | | 0.007* | | | |
| IL_17A | 2.19 + 1.65 | 1.466 + 0.98 | 2.63 + 2.62 | 1.51 + 0.74 | 0.473 | 0.199 |
| p-value | 0.412 | | 0.167 | | | |
| IL_18 | 407.453 + 191.381 | 299.355 + 272.716 | 377.6 + 205.65 | 324.88 + 209.21 | 0.706 | 0.828 |
| p-value | 0.345 | | 0.549 | | | |
| IL_1Beta | 83.573 + 76.521 | 26.112 + 26.986 | 114.21 + 103.65 | 52.19 + 51.88 | 0.539 | 0.269 |
| p-value | 0.035* | | 0.080 | | | |

**Note:**
* Significant level at $p < 0.05$.

**Table 5 Spearman's correlation between clinical, salivary and serological parameters.**

| Measurements | Spearman's rho | EMA | Anti_gliadin_Ab | tTG IgA | tTG A |
|---|---|---|---|---|---|
| PI Score | r | −0.245 | −0.042 | −0.520* | −0.14 |
| | p-value | 0.26 | 0.848 | 0.011 | 0.524 |
| GI Score | r | −0.151 | −0.005 | −0.584** | −0.179 |
| | p-value | 0.492 | 0.981 | 0.003 | 0.413 |
| BOP% | r | −0.232 | −0.057 | −0.594** | −0.267 |
| | p-value | 0.287 | 0.797 | 0.003 | 0.218 |
| IL_17A µdl | r | −0.22 | −0.039 | −0.144 | −0.247 |
| | p-value | 0.312 | 0.861 | 0.513 | 0.256 |
| IL_18 µdl | r | −0.026 | 0.013 | −0.128 | 0.03 |
| | p-value | 0.907 | 0.954 | 0.561 | 0.891 |
| IL_1Beta µdl | r | 0.073 | 0.168 | −0.083 | 0.141 |
| | p-value | 0.739 | 0.442 | 0.708 | 0.52 |

**Notes:**
** Correlation is significant at $p < 0.01$ level.
* Correlation is significant at $p < 0.05$ level.

# DISCUSSION

In this study, the salivary levels of interleukin-17A, interleukin-1beta, and interleukin-18 in 23 celiac disease patients on a gluten-free diet were evaluated against 23 non-celiac controls. The drooling method is noted for its simplicity and high acceptance, particularly among certain groups such as: children, handicapped and un co-operative patients (*Bhattarai, Kim & Chae, 2018*). The study selected for the spitting method over drooling for saliva collection due to its capacity to gather both stimulated and unstimulated saliva,

**Table 6 Possible association of each biomarker with disease (control vs celiac).**

| Biomarkers variables celiac vs control | B | Std. Error | t | Sig. | 95.0% confidence interval for B | |
|---|---|---|---|---|---|---|
| | | | | | Lower bound | Upper bound |
| GI* | −0.475 | 0.126 | −3.771 | 0.001 | −0.729 | −0.221 |
| min_PD* | −1.267 | 0.198 | −6.41 | 0.000 | −1.666 | −0.868 |
| max_PD* | −1.524 | 0.269 | −5.675 | 0.000 | −2.066 | −0.982 |
| min_CAL* | −1.311 | 0.245 | −5.359 | 0.000 | −1.804 | −0.817 |
| max_CAL* | −2.799 | 0.453 | −6.176 | 0.000 | −3.714 | −1.884 |
| BOP* | −0.183 | 0.043 | −4.242 | 0.000 | −0.27 | −0.096 |
| IL_17A | −0.046 | 0.516 | −0.09 | 0.929 | −1.088 | 0.995 |
| IL_18 | −54.91 | 61.402 | −0.894 | 0.376 | −178.657 | 68.837 |
| IL_1Beta | −4.765 | 24.864 | −0.192 | 0.849 | −54.875 | 45.345 |

**Note:**
* Linear regression model was statistically significant at 0.05 level of significance.

reflecting a comprehensive oral health status. This method is known for its robustness in measuring salivary flow rate and content, fundamental to our study's investigation (*Fey, Bikker & Hesse, 2024*; *Özçaka, Nalbantsoy & Buduneli, 2011*).

Recognizing periodontal diseases risk factors like dental hygiene, diabetes, and smoking history is crucial (*Axelsson, Nyström & Lindhe, 2004*; *Grössner-Schreiber et al., 2006*; *Madi et al., 2023*); hence, in this study exclusion of smokers and diabetic subjects was conducted to avoid the effect of these factors on the study findings. In the current study, the clinical periodontal parameters (PI, GI, BOP, PD, CAL) were notably higher in non-celiac subjects. In the control group, these parameters significantly escalated in patients with periodontitis. In the celiac cohort, similar trends were observed, but without statistical significance, possibly due to the small subgroup size and heightened awareness of diet and oral hygiene practice in celiac patients. The periodontitis patients in the control group showed significantly higher GI and BOP than periodontitis patient with celiac diseases, this could be due to difference in number of patients in both subgroups. The relationship between periodontal health and systemic conditions like inflammatory bowel diseases (IBD) showed diverse findings. *Brito et al. (2013)*, observed no significant differences in periodontal parameters between inflammatory bowel diseases IBD patients and healthy controls with periodontitis. This was further supported by *Tan et al. (2021)* and *Koutsochristou et al. (2015)* reporting no notable differences in plaque scores among Greek adolescents. Contrasting with our findings, *Grössner-Schreiber et al. (2006)* and *Johannsen et al. (2015)* reported no significant differences in periodontal disease parameters between IBD and non-IBD patients.

*Buchbender et al. (2022)*, observed no significant difference in Mombelli plaque index (mPI) between the control (HC) and Crohn's disease (CD) patients. However, the current data indicated a higher mean BOP value in the CD group compared to the control group. This aligns with increasing evidence linking IBD with periodontal disease due to similar mucosal responses to microorganisms (*Agossa et al., 2017*; *Papageorgiou et al., 2017*; *Zhang et al., 2020*).

*Figueredo et al. (2008)* examined gingival crevicular fluid in IBD and periodontitis patients, analysing cytokines like IL-4, -6, -1ß, -18, and IFN-γ, but found no significant differences in them or in periodontal clinical parameters (BOP, CAL, PI). *Noh et al. (2013)* identified IL-6 in inflamed gingival tissue, while *Buchbender et al. (2022)*, detected it across all groups without notable variations, suggesting no direct link between IBD and periodontitis.

Dietary factors, such as high sugar (*Beklen et al., 2007*) and carbohydrate intake (*Nascimento et al., 2017*), have been associated with periodontal disease, highlighting the role of diet in periodontal health. *Wright et al. (2020)* observed that a healthy diet that was rich in fruits, and vegetables led to reduced attachment loss in American adults. On the other hand, *Alhassani et al. (2021)* linked ultra-processed diet consumption with an increased risk of periodontitis.

The current study found no significant difference in salivary IL-17A levels between control and CD groups, though control subjects with periodontitis showed higher levels. The role of IL-17A in periodontal disease, as indicated by *Techatanawat et al. (2020)*, points to its involvement in chronic inflammation. Contrasting findings on IL-17A and IL-18 levels in periodontal contexts were reported by previous researchers. While clinical studies have noted elevated IL-17 levels in in GCF, saliva, and serum of periodontitis patients (*Azman et al., 2014*; *Ohyama et al., 2009*; *Okui et al., 2012*; *Takahashi et al., 2005*), other studies (*Ay et al., 2012*; *Johannsen et al., 2015*; *Özçaka, Nalbantsoy & Buduneli, 2011*; *Vahabi, Yadegari & Pournaghi, 2020*) observed lowered salivary IL-17 in periodontitis patient than control.

While the precise role of IL-17 in periodontal disease development and host defence remains unclear (*Yu & Gaffen, 2008*). Some suggest that IL-17's role in periodontitis could be more localized rather than systemic, which could be why no significant differences were observed in our study. This aligns with the complex and multifactorial nature of periodontitis and celiac disease interaction (*Vahabi, Yadegari & Pournaghi, 2020*). The specific dynamics between these conditions and their impact on IL-17A levels could be an intricate part of this interrelation. Additionally, diseased gingival tissues exhibit increased IL-17 mRNA (*Cardoso et al., 2009*; *Cardoso, Reis & Manzanares-Céspedes, 2018*), and periodontitis lesions, particularly progressive ones, show higher IL-17 and RANKL mRNA expression compared to inactive lesions, indicating a potential link between Th17 cytokines, bone resorption, and periodontal disease activity (*Özçaka, Nalbantsoy & Buduneli, 2011*).

In the current study, the level of IL-18 was higher in periodontitis patients than healthy patients in both the control and celiac groups. Similarly, elevated IL-18 levels in saliva (*Özçaka, Nalbantsoy & Buduneli, 2011*) and GCF (*Alhassani et al., 2021*; *Pradeep et al., 2009*) of periodontitis patients were observed. Elevated Salivary IL-18 level was suggested as a biomarker for periodontal tissue destruction (*Özçaka, Nalbantsoy & Buduneli, 2011*). *Figueredo et al. (2008)* and *Orozco et al. (2006)* reported higher IL-18 levels at inflamed periodontal sites regardless of pocket depth, indicating its potential role in periodontitis. However, *Esfahrood et al. (2016)* observed no significant difference in IL-18 levels between healthy and periodontitis patients, while *Sánchez-Hernández et al. (2011)* found elevated serum IL-18 in periodontitis patients. These findings suggest a complex involvement of IL-

18 in periodontitis, warranting further investigation. Furthermore, education about celiac disease's aetiology and dietary adjustments, as demonstrated by *Spinell et al. (2018)*, can positively affect clinical outcomes. A gluten-free diet may attenuate systemic inflammation, influencing cytokines crucial in periodontal diseases (*Spinell et al., 2018*; *Volta et al., 2012*). This could explain the current findings that the decreased interleukin levels could be due to the gluten-free diet and that this influences the decrease in periodontal signs as well. Previous research indicates a decline in antiGA IgG levels among non-celiac gluten sensitivity patients adhering to a gluten-free diet, while celiac patients exhibit a 60% reduction (*Volta et al., 2012*).

This diet-induced modulation of inflammatory pathways could contribute to the observed periodontal health benefits in celiac patients. Further investigation and direct comparisons in clinical studies would be essential to establish a definitive link between these dietary practices, cytokine levels, and periodontal health outcomes. The relationship between a gluten-free diet and periodontal health in celiac disease patients is multifaceted. *Cervino et al. (2018)* suggest that a gluten-free diet positively affects oral health in celiac patients. *Potter et al. (2018)* review the diet's broader health impacts, including potential influences on cardiovascular and periodontal health. *Bascuñán, Vespa & Araya (2017)* described the gluten-free diet's complexities, highlighting its strict requirements and potential impact on overall health, including periodontal wellbeing. These studies provide a starting point for understanding the potential links between a gluten-free diet, its broader health implications, and specifically its impact on periodontal health in celiac disease patients.

The fact that no significant changes were detected for patients with CD in this study could be attributed to the small sample size and unbalanced group, suggesting the need for a larger, multi-centre study on the impact of a gluten-free diet (GFD) on periodontal health. The study limitations include the inability to generalize results due to the small sample size, the lack of matching between study groups, and the omission of oral bacterial composition in the analysis.

## CONCLUSIONS

The study indicates that celiac patients on gluten-free diet exhibited better periodontal health compared to non-celiac patients, as reflected in the lower periodontal parameters. While salivary biomarkers showed no significant difference between groups, the control group displayed higher levels of IL-1Beta in periodontitis patients. Correlation analysis revealed an inverse relationship between periodontal parameters and IgA, underscoring the complex interplay between diet, immune response, and periodontal health in celiac disease.

### Funding

The authors were supported by the Deputyship of Research and Innovation, Ministry of Education, Kingdom of Saudi Arabi, by funding through the project number IF-2020-006-Dent, at the Imam Abdulrahman Bin Faisal University, College of Dentistry. The funders

had no role in study design, data collection and analysis, decision to publish, or preparation of the manuscript.

## Grant Disclosures
The following grant information was disclosed by the authors:
Ministry of Education: IF-2020-006-Dent.
Imam Abdulrahman Bin Faisal University, College of Dentistry.

## Competing Interests
The authors declare that they have no competing interests.

## Author Contributions
- Marwa Madi conceived and designed the experiments, performed the experiments, analyzed the data, prepared figures and/or tables, authored or reviewed drafts of the article, and approved the final draft.
- Maha Abdelsalam conceived and designed the experiments, performed the experiments, analyzed the data, prepared figures and/or tables, authored or reviewed drafts of the article, and approved the final draft.
- Ahmed Elakel conceived and designed the experiments, performed the experiments, analyzed the data, prepared figures and/or tables, authored or reviewed drafts of the article, and approved the final draft.
- Osama Zakaria conceived and designed the experiments, performed the experiments, analyzed the data, prepared figures and/or tables, authored or reviewed drafts of the article, and approved the final draft.
- Maher AlGhamdi performed the experiments, analyzed the data, prepared figures and/or tables, authored or reviewed drafts of the article, and approved the final draft.
- Mohammed Alqahtani performed the experiments, analyzed the data, prepared figures and/or tables, authored or reviewed drafts of the article, and approved the final draft.
- Luba AlMuhaish performed the experiments, analyzed the data, prepared figures and/or tables, authored or reviewed drafts of the article, and approved the final draft.
- Faraz Farooqi performed the experiments, analyzed the data, prepared figures and/or tables, authored or reviewed drafts of the article, and approved the final draft.
- Turki A. Alamri performed the experiments, analyzed the data, prepared figures and/or tables, authored or reviewed drafts of the article, and approved the final draft.
- Ibrahim A. Alhafid performed the experiments, analyzed the data, prepared figures and/or tables, authored or reviewed drafts of the article, and approved the final draft.
- Ibrahim M. Alzahrani performed the experiments, analyzed the data, prepared figures and/or tables, authored or reviewed drafts of the article, and approved the final draft.
- Adel H. Alam conceived and designed the experiments, performed the experiments, analyzed the data, prepared figures and/or tables, authored or reviewed drafts of the article, and approved the final draft.
- Majed T. Alhashmi performed the experiments, analyzed the data, prepared figures and/or tables, authored or reviewed drafts of the article, and approved the final draft.

- Ibrahim A. Alasseri performed the experiments, analyzed the data, prepared figures and/or tables, authored or reviewed drafts of the article, and approved the final draft.
- Ahmad A. AlQuorain performed the experiments, analyzed the data, prepared figures and/or tables, authored or reviewed drafts of the article, and approved the final draft.
- Abdulaziz A. AlQuorain conceived and designed the experiments, performed the experiments, analyzed the data, prepared figures and/or tables, authored or reviewed drafts of the article, and approved the final draft.

### Human Ethics

The following information was supplied relating to ethical approvals (*i.e.*, approving body and any reference numbers):

The Institutional Review Board of Imam Abdulrahman Bin Faisal University approved the study proposal (IRB-2021-02-034).

### Ethics

The following information was supplied relating to ethical approvals (*i.e.*, approving body and any reference numbers):

The Institutional Review Board of Imam Abdulrahman Bin Faisal University.

### Data Availability

The data is available at figshare: Madi, Marwa (2023). Salivary interleukin-17A and interleukin-18 levels in patients with Celiac disease and periodontitis. figshare. Dataset. https://doi.org/10.6084/m9.figshare.24250018.v2.

### Supplemental Information

Supplemental information for this article can be found online at http://dx.doi.org/10.7717/peerj.17374#supplemental-information.

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
