# Peer review of "Salivary interleukin-17A and interleukin-18 levels in patients with celiac disease and periodontitis"

_PeerJ, doi:10.7717/peerj.17374_

## Round 0.1 · original submission · Major Revisions

Thanks for submitting the manuscript. Please address the queries raised by the authors along with the following comments-

1. Don't start the abstract with 'Previous studies.....'. Start with a statement from the research itself

2. Try to proofread the work with the help of a native English speaker

3. Before conducting the Mann-Whitney, you need to assess the normality of the data. Please mention that.

4. Correct the statistical interpretations.

**Language Note:** The review process has identified that the English language must be improved. PeerJ can provide language editing services - please contact us at [email protected] for pricing (be sure to provide your manuscript number and title). Alternatively, you should make your own arrangements to improve the language quality and provide details in your response letter. – PeerJ Staff

Reviewer 1 ·

Basic reporting

English must be corrected. Most of the literature is current. Raw data is missing. The authors presented the data code, but did not show detailed clinical trial and ELISA results for individual patients.

Experimental design

The reviewed work is interesting. The authors examined the levels of salivary IL-17A, IL-18 and IL-1B in patients with celiac disease and controls. In the study, they correctly used clinical examination and ELISA method for IL-17A, IL-18, and IL-1B measurements. Ultimately, they showed that celiac patients under gluten-free diet had better periodontal health than control subjects. At the same time, increased salivary IL-17A, IL-18 and IL-1B levels were associated with periodontitis, but not with celiac. The authors suggest that anti-inflammatory diet has protective effects on periodontal health.
Unfortunately, there are several shortcomings in the work:
1. Test methodology for serum IgA, tTG antibody [tTGA], and anti-endomysial antibody [EMA IgA] is not presented.
2. Patient groups for periodontitis are not properly grouped. In the control group, 83% of people had periodontitis, and in the celiac disease group only 30% of patients had periodontitis. In both groups, the number of people with periodontitis should be similar. Unfortunately, the number of people (only 4) without periodontitis in the control group is too small for statistical analysis and could have influenced the results. It is possible that this also had an impact on the higher levels of GI, BOP and CAL in the control group. This leads to doubts about the validity of the results.
3. The discussion is very inconsistent, composed of sentences taken out of context, and sometimes it is unclear what the authors wanted to convey.

Validity of the findings

Before publication, the study groups must be corrected to ensure that the number of people with and without periodontitis is similar in both groups.

·

Basic reporting

- The manuscript is generally well-established but it could definitely
benefit from English language editing especially in the results and
discussion sections, lengthy non-linked sentences were used, for
example:
1. line 190 can be improved saying: (comparison of salivary
markers levels between the controls and celiac groups are
presented in table 4).
2. Lines 192,194: using the term “periodontics patients’ is not
correct from a periodontal point of view, as periodontists we
refer to patients with periodontal disease using the term
(periodontal disease patients) and according to the study
subjects, the author could use (periodontitis patients).
3. Line 191: the term “chronic’ is no longer used according to
the classification system of 2017 for periodontal disease and
conditions, the author can include the grade instead to express
the rate of progression.
4. Tables are well put without excessive complexity and are easy
to comprehend.
5. the authors efforts in collecting all relevant studies that are up
to date is obvious.
6. References over 15 years are outdated, if possible, please
consider citing references within 5 years period (line
357,376).
7. The author reported results of periodontal clinical parameters
in line 183 and then again in 196, to avoid any confusion for
the reader, please mention the type of comparison in each
section for example line 196: (intra- group comparison of the
celiac group revealed ……).

Experimental design

- the execution of the study design is good, however one of the main
weaknesses was pointed out by the author which is the non-matched
periodontitis and healthy periodontium subgroups that might have
dramatically impacted the results.
- The aim of the study was ‘to investigate the levels of salivary
interleukin-17A, interleukin -1beta, and interleukin-18 in celiac
disease patients under gluten-free diet with and without periodontitis
and compare them with healthy controls”, yet there wasn’t any
comparison with regard to biomarkers levels and periodontal
parameters between healthy periodontium of controls and those with
celiac disease, and the similar applies for the periodontitis,
preferably the author should point it as another limitation.
- According to Bhattarai et al (Int J Med Sci, 2018), passive drooling
is preferred in terms of salivary composition, can the author explain
the reason for choosing the spitting method?
- The methodology is well described with the sufficient details.

Validity of the findings

- Since the control group had the greatest share of periodontitis cases,
isn’t it obvious that they are going to end up having the highest values
in all clinical parameters?
- in the conclusion, the author reported “Celiac disease patients under
gluten-free diet had better periodontal health than control” again, no
comparison was conducted between non-celiac periodontally healthy
and celiac with healthy periodontium, please consider re-writing the
conclusion.
- In order to examine the association and the effect of the disease with
the selected biomarkers which is the main theme of the study, multiple
regression models should be conducted in each group, this is another
limitation of the study.

Reviewer 3 ·

Basic reporting

Introduction

1. I consider that the concept of etiology of periodontitis and its description can be supported by the more modern concept of dysbiosis. I attach two references that can be used (lanes 73-79)

a) Kinane DF, Stathopoulou PG, Papapanou PN. Periodontal diseases. Nat Rev Dis Primers. 2017 Jun 22;3:17038. doi: 10.1038/nrdp.2017.38. PMID: 28805207.

b) Hajishengallis G, Chavakis T. Local and systemic mechanisms linking periodontal disease and inflammatory comorbidities. Nat Rev Immunol. 2021 Jul;21(7):426-440. doi: 10.1038/s41577-020-00488-6. Epub 2021 Jan 28. PMID: 33510490; PMCID: PMC7841384.

2. It is described that it is important to evaluate IL-1beta, however, it is not clear what role it plays in celiac disease and periodontitis.

Results

3. Considering that no significant differences were found in the levels of IL-1 beta, IL-17A and IL-18 between the control group and the group with celiac disease, it should not be mentioned that there were higher levels in the control group. Because mathematically this difference does not exist (lanes 186-188).

4. The non-significant P value is expressed as P>0.05 (lane 188)

5. In general, the results should express in which units the variables are being reported and adequately describe how the results are being reported (means and standard deviations or medians).

Discussion

6. I consider that from line 255 the sentence should begin like “On the other hand…. “An association… (lanes 255 and 256).

7. I consider that instead of emphasizing that no significant differences in IL-17A were found between the groups, it is better to support why no statistical differences were observed, supported by the reference of Özcaka 2011 and Vahabi S (2020), for periodontitis, and since relate this point to celiac disease (lanes 273-277).

a) Vahabi S, Yadegari Z, Pournaghi S. The comparison of the salivary concentration of interleukin-17 and interleukin-18 in patients with chronic periodontitis and healthy individuals. Dent Res J (Isfahan). 2020 Aug 14;17(4):280-286. PMID: 33282154; PMCID: PMC7688042.

8. Regarding IL-18 in serum in patients with periodontitis, you can refer to this paper (lanes 284-290)

b) Sánchez-Hernández PE, Zamora-Perez AL, Fuentes-Lerma M, Robles-Gómez C, Mariaud-Schmidt RP, Guerrero-Velázquez C. IL-12 and IL-18 levels in serum and gingival tissue in aggressive and chronic periodontitis. Oral Dis. 2011 Jul;17(5):522-9. doi: 10.1111/j.1601-0825.2011.01798.x. Epub 2011 Feb 18. PMID: 21332601.

9. Y de IL-18 en líquido crevicular gingival con las siguientes referencias

a) Pradeep AR, Hadge P, Chowdhry S, Patel S, Happy D. Exploring the role of Th1 cytokines: interleukin-17 and interleukin-18 in periodontal health and disease. J Oral Sci. 2009 Jun;51(2):261-6. doi: 10.2334/josnusd.51.261. PMID: 19550095.

10. Finally, I consider that the most important point to discuss is to elucidate why patients with celiac disease on a gluten-free diet have better periodontal health and this to begin to relate it to the levels of IL-1 beta, IL-17A and IL-18 reported for celiac disease by other authors. At the same time, I consider it important to include a group of patients diagnosed with periodontitis, to consider that, regardless of the small sample size, the non-significant differences in the interleukins studied are due to the gluten-free diet and that this influences the decrease in periodontal signs.

Conclusions

11. Delete conclusion from IL-17A and IL-18, because the significant increase in periodontitis was only observed in IL-1 beta.

Tables
Table 2. Add units of pedriodontal parameters (mm or %)

Experimental design

no comment

Validity of the findings

no comment

---

## Round 0.2 · accepted · Accept

Thanks for making all the necessary changes.

·

Basic reporting

No comment

Experimental design

no comment

Validity of the findings

no more comments

Additional comments

the authors have done a pretty good job in responding to reviewers, the edited sections are well put and connected.

Reviewer 3 ·

Basic reporting

The article has been satisfactorily modified according to the suggestions.
The recommendation is that it be accepted

Experimental design

None

Validity of the findings

None

Additional comments

None